# Processed and ultra-processed foods are associated with high prevalence of inadequate selenium intake and low prevalence of vitamin B1 and zinc inadequacy in adolescents from public schools in an urban area of northeastern Brazil

**Raphaela Cecília Thé Maia de Arruda Falcão**[1☯], **Clélia de Oliveira Lyra**[1,2☯], **Célia Márcia Medeiros de Morais**[2☯], **Liana Galvão Bacurau Pinheiro**[2☯], **Lucia Fátima Campos Pedrosa**[1,2☯], **Severina Carla Vieira Cunha Lima**[1,2‡], **Karine Cavalcanti Maurício Sena-Evangelista**[1,2‡]*

**1** Postgraduate Nutrition Program, Center for Health Sciences, Federal University of Rio Grande do Norte, Natal, Brazil, **2** Department of Nutrition, Federal University of Rio Grande do Norte, Natal, Brazil

☯ These authors contributed equally to this work.
‡ These authors also contributed equally to this work.
* kcmsena@gmail.com

## Abstract

Changes in eating behavior of adolescents are associated with high consumption of processed and ultra-processed foods. This study evaluated the association between these foods and the prevalence of inadequate micronutrient intake in adolescents. A cross-sectional study was conducted with 444 adolescents from public schools in the city of Natal, northeastern Brazil. The adolescents' habitual food consumption was evaluated using two 24-hour dietary recalls. Foods were categorized according to the degree of processing (processed and ultra-processed) and distributed into energy quartiles, using the NOVA classification system. Inadequacies in micronutrient intake were assessed using the estimated average requirement (EAR) as the cutoff point. Multivariate logistic regression models were used to estimate the relationship between energy percentage from processed and ultra-processed foods and prevalence of inadequate micronutrient intake. The mean (Standard Deviation (SD)) consumption of total energy from processed foods ranged from 5.8% (1.7%) in Q1 to 20.6% (2.9%) in Q4, while the mean consumption of total energy from ultra-processed foods ranged from 21.4% (4.9%) in Q1 to 61.5% (11.7%) in Q4. The rates of inadequate intake of vitamin D, vitamin E, folate, calcium, and selenium were above 80% for both sexes across all age groups. Energy consumption from processed foods was associated with higher prevalence of inadequate selenium intake (p < 0.01) and lower prevalence of inadequate vitamin B1 intake (p = 0.04). Energy consumption from ultra-processed foods was associated with lower prevalence of inadequate zinc and vitamin B1 intake (p < 0.01 and p = 0.03, respectively). An increase in the proportion of energy obtained from processed and

**Data Availability Statement:** All relevant data are within the paper and Harvard Dataverse repository (https://doi.org/10.7910/DVN/JUKPWQ).

**Funding:** This research was supported by the National Council for Scientific and Technological Development (Conselho Nacional de Desenvolvimento Científico e Tecnológico – CNPq, Brazil; grant no. 478287-06-2) and the Research Support Foundation of Rio Grande do Norte (Fundação de Apoio à Pesquisa do Rio Grande do Norte – FAPERN; Brazil)/Coordination for the Improvement of Higher Education Personnel (Coordenação de Aperfeiçoamento de Pessoal de Nível Superior – CAPES, Brazil - 006/2014. This study was also financed in part by the Coordenação de Aperfeiçoamento de Pessoal de Nível Superior – Brasil (CAPES) – Finance Code 001.

**Competing interests:** The authors have declared that no competing interests exist.

ultra-processed foods may reflect higher prevalence of inadequate selenium intake and lower prevalence of vitamin B1 and zinc inadequacy.

## Introduction

Adolescence is a period of high nutrient and energy demand and is thus a nutritionally critical period of life when lifestyle and dietary habits are changing [1]. This makes adolescents vulnerable to the consumption of heavily-processed, high-fat and high-sugar products [2, 3].

To study the effect of food processing on nutritional quality and health, we used the recently developed NOVA (which is not an acronym) classification system, which has been recognized as an appropriate framework for assessment of food processing levels [4]. NOVA is a system of grouping foods according to the nature, extent and purpose of the industrial processing used in their production. According to this system, all foods and beverages can be categorized into four groups: (i) unprocessed or minimally processed foods, (ii) processed culinary ingredients, (iii) processed foods, and (iv) ultra-processed foods. Processed foods are products manufactured by adding sugar, oil, salt, and other culinary ingredients to minimally processed foods to make them more durable and usually more palatable, for example, French bread, cheese, processed meats, canned fruits, and vegetables. Ultra-processed foods differ from less processed foods in that they fail to maintain their basic identity, undergoing various processing stages and techniques and including substances used exclusively in industry, such as cakes, pies, cookies, ready-to-eat and semi-ready-to-eat meals, bakery products, sugar-sweetened beverages, and snacks [4].

Meanwhile, among adolescents, some authors stress that the time spent in sedentary behaviors can lead to greater consumption of foods purchased in packaged form, ready to eat or heat [3]. The high consumption of processed and ultra-processed foods among adolescents has been associated with higher energy, saturated fat, or trans-fat intake, and a lower intake of dietary fiber and micronutrients [5, 6]. Inadequate food consumption patterns during childhood and adolescence are linked not only with the occurrence of obesity in youth, but also with the subsequent risk of developing diabetes, hypertension, and other noncommunicable diseases in adulthood [7]. In a healthy population of urban European adolescents, the HELENA (Healthy Lifestyle in Europe by Nutrition in Adolescence) Study showed that adolescents eat half of the recommended amount of fruit and vegetables and less than two-thirds of the recommended amount of milk (and milk products), but consume much more meat (and meat products), fats, and sweets than recommended [8].

In the United States (US), using 2009–2010 National Health and Nutrition Examination Survey (NHANES) data, an association was reported between consumption of ultra-processed foods and dietary content of added sugars [9]. Furthermore, the average content of protein, fiber, vitamins A, C, D, and E, zinc, potassium, phosphorus, magnesium, and calcium in the diet decreased significantly with an increase in the energy contribution of ultra-processed food [10]. Another study found that 7 to 20% of children and adolescents in the US miss lunch on both weekdays and weekends, and this practice was associated with lower micronutrient intake [11]. A systematic review points out that adolescents from Malaysia, a middle-income country in Asia, had lower diet quality, and Chinese adolescents spent less time in physical activity compared to other ethnicities [12]. These dietary patterns in adolescence are predicted to become global trends, and they have been associated with health risks for chronic diseases in

adulthood, as this critical period of development forms the basis for establishing a lifetime of eating habits [13].

Latin American countries have experienced major shifts in intake of less-healthful low-nutrient-density foods and sugary beverages, changes in away-from-home eating and snacking, and rapid shifts towards high levels of overweight and obesity among all ages [14]. In a study of children from Colombia, there was a decline in intake of n-3 polyunsaturated fatty acids, calcium, zinc, and vitamins A, B12, C, and E, and this lower intake was related to an increased intake of processed and ultra-processed foods, as well as higher sodium, sugar, and trans-fatty acids intake. This study also observed that the amounts of some healthy nutrients, including folate and iron, were higher in processed and ultra-processed foods than that in unprocessed and minimally processed foods [15].

Brazilian household surveys conducted in the past three decades have shown a steady increase in ultra-processed food intake and a significant reduction in the intake of natural or minimally processed foods in all geographical regions [16, 17]. The 2013–2014 Brazilian Study of Cardiovascular Risks in Adolescents (ERICA) study of 71,791 adolescents aged from 12 to 17 years showed that rice, beans and other legumes, juice and fruit drinks, breads, and meat were the most commonly consumed foods. Saturated fat and free sugar intake were above the maximum recommended limit ($< 10.0\%$). In short, the diets of Brazilian adolescents were characterized by the intake of traditional Brazilian food, such as rice and beans, as well as by high intake of sugar through sweetened beverages and processed foods. This food pattern was associated with an excessive intake of sodium, saturated fatty acids, and free sugar [18]. In addition, the Brazilian government National Adolescent School-based Health Survey (PeNSE 2015) points out that adolescents from the northeastern region of Brazil, whose mothers have a lower level of schooling, have a healthier dietary pattern compared to other more developed regions of Brazil [19,20].

To date, there are gaps in the literature of studies that focused on the association between intake of processed and ultra-processed foods and inadequate intake of micronutrients in adolescents. Only one study conducted among young people from southern Brazil reported that a higher consumption of ultra-processed foods was associated with a higher prevalence of inadequate intake of calcium, sodium, and iron [6]. Therefore, the purpose of this study was to evaluate the association of consumption of processed and ultra-processed foods with the prevalence of inadequate micronutrient intake in adolescents from public schools in an urban area of northeastern Brazil, given the urgent need to formulate dietary recommendations that favor the promotion of healthy eating.

## Methods

### Study population

This cross-sectional study was conducted among 444 adolescents from public schools in the city of Natal, northeastern Brazil. Data were collected in 2007 and 2008 as part of the "Risk factors for cardiovascular disease among adolescent beneficiaries of the Brazilian government National School Lunch Program (PNAE)" study [21–23].

Inclusion criteria corresponded to adolescents' regular attendance at schools and the age group of interest (10 to 19 years) according to the date of birth recorded in the school's enrollment document. Exclusion criteria corresponded to the presence of genetic syndromes associated with obesity (Down syndrome and muscular dystrophies), cerebral palsy, adolescent pregnancy, and adolescents with special needs or who used medications such as corticosteroids that could alter the results of biochemical tests. Adolescents were previously evaluated by a

team of endocrinologists participating in the study, who identified the presence of these health conditions that excluded study participation.

The sampling plan was defined using a two-stage random stratified sampling strategy based on a population of 39,920 elementary and high school students from 66 municipal schools, and considering the four sanitary districts of the city: north, n = 19,270; south, n = 4,128; west, n = 3,728; and east, n = 12,794. A pilot study was developed in four schools, one in each district of the city, with the purpose of estimating the prevalence of altered lipid profiles in each district. A maximum error of 4% was considered, which resulted in a sample size of 483 adolescents. Stratified sampling using Neyman allocation was performed to define the sample size in each district, as follows: north, n = 285; south, n = 63; west, n = 34; and east, n = 101. However, 711 students were enrolled to compensate for losses of 30% in the sample (which corresponds to 628 students). To determine the number of schools, the average number of students per school was considered, assuming that the variance of that number in the four districts was approximately equal. The school sample size obtained was n = 21, and according to proportional allocation, the following numbers of schools were obtained for each stratum: north, n = 9; south, n = 3; east, n = 3; and west, n = 6.

The selection of schools was conducted by systematic draw, with the sample of students distributed randomly by district and in a manner proportional to the total number of students drawn from the schools. The sample units were selected from the student list, which was used to unify all series with sequential numbering and a systematic draw, without replacement, in case of refusal of the selected participants. For quality control and to reduce sample losses, there were up to two school visits when the number of absentees was higher than 30%. Fig 1 shows the number of adolescents eligible, adolescents enrolled, and details of sample losses. Of the 450 adolescents with complete information on dietary intake, data from six adolescents were excluded because of reported energy intakes <500 and >5000 kcal [24], resulting in a final population of 444 adolescents.

## Ethical approval

The study was approved by the Research Ethics Committee of the Federal University of Rio Grande do Norte (No. 112/06). All adolescents and their parents or legal guardians provided informed consent for inclusion before participating in the study. The study is not a clinical trial and, therefore, it does not need to be registered.

## Anthropometric assessment and sexual maturation

Weight and height were measured for calculation of body mass index (BMI), which was then classified according to World Health Organization criteria (2007) [25]. All anthropometric measurements (weight and height) were performed at the school in duplicate by trained staff, according to Habicht [26] standardization techniques. An analysis of the reproducibility of the anthropometric measurements showed no difference between the means of the first and second measurements of each trained evaluator. A medical team of pediatric endocrinologists assessed pubertal staging. Sexual maturation was classified according to the Tanner scale into: pre-pubertal (stage 1), initial pubertal (stages 2 and 3), and final pubertal (stages 4 and 5) [27, 28].

## Food intake assessment

Food intake data were obtained using two 24-hour dietary recalls administered by trained staff to all study participants at an interval of 30–45 days. Recall surveys were performed according to the recommendation of Thompson and Byers [29], with emphasis on the following criteria:

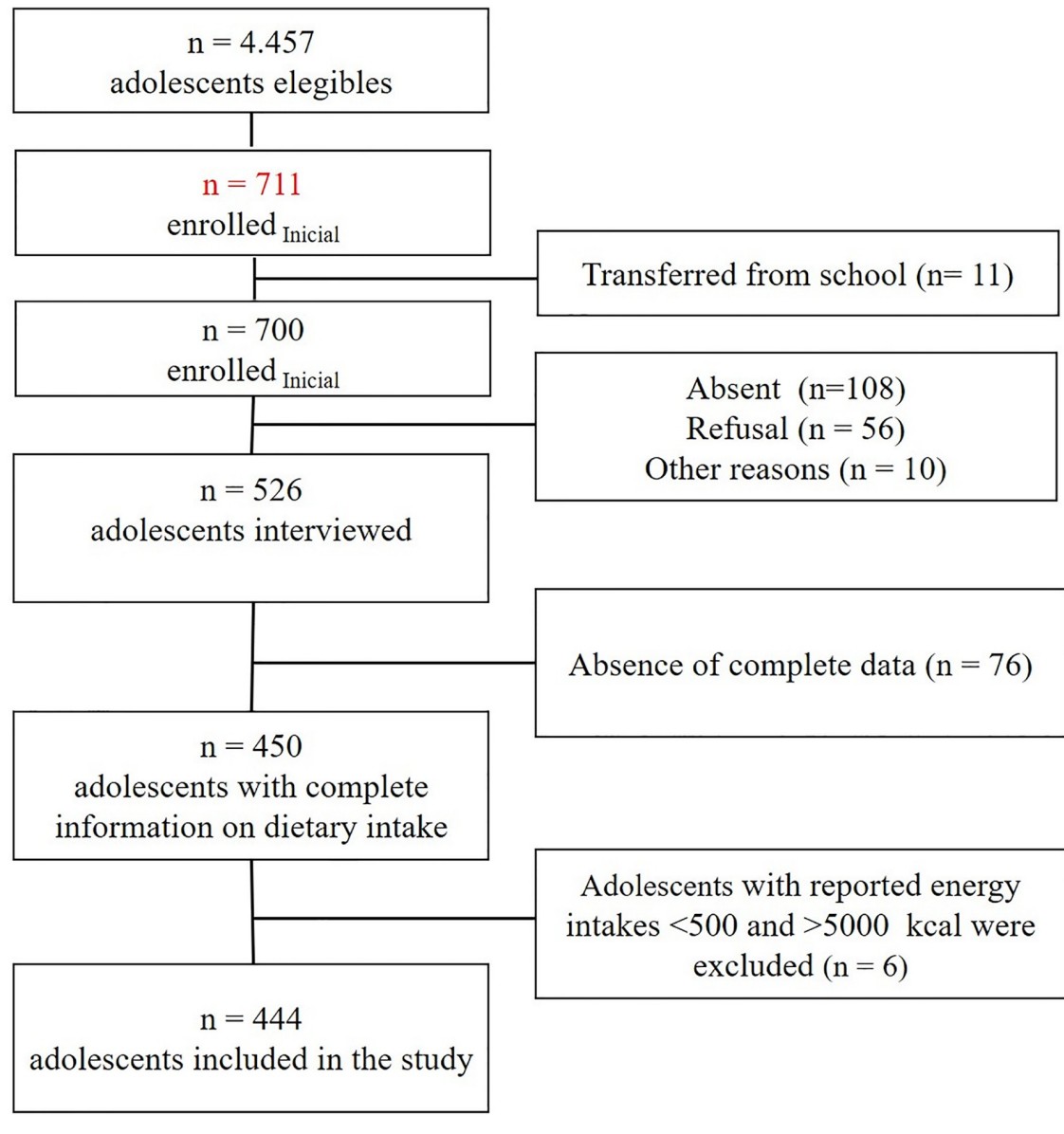

**Fig 1. Flowchart of participant selection.**

(1) alternate days, except Mondays, avoiding atypical data; (2) distinct periods in each month, considering the purchasing power of families; and (3) different periods of the year, considering the seasonal availability of food. Photographs of utensils and containers were used to identify food-serving items and quantify serving sizes, classified as small, average, or large. The quantity of each food or drink was converted into grams or milliliters using a measurement table that refers to food consumed in Brazil [30]. The foods were converted into energy and nutrients using the software Virtual Nutri Plus® 2.0 (São Paulo, Brazil) [31]. New preparations and foods were added to the software database as necessary, along with their nutritional composition, from the Brazilian food composition table [32] and the United States Department of Agriculture (USDA) [33] database, as appropriate. Nutritional information collected from industrial food labels was also added to the software database.

The nutritional information from both 24-hour recalls were inserted in the Multiple Source Method (MSM) https://msm.dife.de/ [34] to calculate usual dietary intake for individuals and then construct the population distribution based on these data. This method was used to correct dietary data for intra-personal variability in each group of processed and ultra-processed foods. Through the MSM, it is possible to estimate the habitual consumption of healthy individuals, using the probability of consumption and the quantity consumed, or the combination of both estimates, removing measurement error from the data [35].

## Food classification according to NOVA and calculation of the prevalence of inadequate micronutrient intake

Foods consumed were classified according to extent and grade of processing using the NOVA classification system proposed by Monteiro et al [4]. This system categorizes food into four groups: (i) unprocessed or minimally processed foods, (ii) processed culinary ingredients designed to be combined with foods to make meals and dishes, (iii) processed foods, and (iv) ultra-processed foods [4,36]. Processed products are foods that have been altered to add substances such as salt, sugar, or oil that substantially change their nature or use. Ultra-processed products are food products formulated mainly or entirely from processed ingredients, typically including little or no whole food. They are durable, edible, drinkable, and palatable by themselves, and are made to be ready-to-consume or ready-to-heat. For the purposes of this study, only the third and fourth groups (processed and ultra-processed products) were analyzed. The foods were then divided into groups as described by Louzada et al [37]. (S1 Table).

Mean and standard deviation for the contribution of each processed or ultra-processed food group to the total energy intake in kilocalories (kcal) were then calculated as a percentage of total energy for each adolescent stratum, corresponding to quartiles of the distribution. The prevalence of inadequate micronutrient intake was estimated using the estimated average requirement (EAR) as the cutoff point for adequate/inadequate intake [38]. The prevalence of inadequate iron intake was calculated using the manually determined probabilistic approach method [39]. The percentage of individuals whose sodium intake exceeded the tolerable upper intake level (UL) was also calculated [40].

## Statistical analysis

A descriptive analysis of the data was performed, and the data were presented as a frequency distribution and 95% confidence interval. The percentages of energy obtained from processed and ultra-processed foods in relation to total energy were transformed into ordinal quantitative variables, with cutoff points based on quartiles of the distribution. A dichotomous outcome variable was generated using the EAR cutoff point, according to sex and age group. for each micronutrient from processed and ultra-processed foods, with subjects categorized as having either adequate or inadequate intake. Inferential analysis to estimate the relationship between the energy percentage from processed and ultra-processed foods and the prevalence of inadequate micronutrient intake was developed using generalized linear models with an ordinal logistic distribution for the outcome variable. Initially, unadjusted models were constructed in a bivariate analysis to identify the association between consumption of processed and ultra-processed foods with the prevalence of inadequate micronutrient intake. Variables with p-values ≤ 0.30 in the unadjusted analysis were selected for inclusion in the adjusted model. The selected variables were then inserted into the model with adjustment using the enter method (all covariates entered into the model in a single step), with overall adjusted significance determined by the p-value associated with the omnibus test of the fitted adjusted model. The significance of each odds ratio (OR) was tested using a Wald chi-squared test, and

the 95% confidence interval was constructed for both the unadjusted and adjusted OR. The estimated measure of effect (OR) was transformed to a prevalence ratio (PR) using the formula $PR = OR(1-P_1) / (1-P_0)$, since PR is more appropriate than OR in cross-sectional studies. This methodological choice was made considering that consumption may not interfere alone in only one micronutrient, and this effect needs to be controlled. A significance level of 0.05 was adopted to minimize type I errors in the analysis of the unadjusted, adjusted, and PR models. The software package SPSS Statistics® version 22.0 was used for data storage and analysis.

## Results

The adolescents had a mean age of 11.8 years. 36.9% of subjects were classified as pre-pubertal and 25.5% as initial puberty. 21.2% of subjects were overweight or obese, and 45.1% of subject mothers were illiterate or had only attended primary school. The general characteristics of the adolescents are shown in Table 1.

The mean consumption of processed foods ranged from 5.8% (1.7%) of the total energy in Q1 to 20.6% (2.9%) in Q4, while the mean consumption of ultra-processed foods ranged from 21.4% (4.9%) of total energy in Q1 to 61.5% (11.7%) in Q4. The most consumed group of processed foods was French bread, whereas the most consumed ultra-processed foods were cakes, pies, and cookies (Table 2).

One hundred percent of both sexes aged 10–13 years displayed inadequate intake of vitamin D, folate, vitamin E, and calcium. More than 80% of adolescents aged 10–13 years had an inadequate intake of phosphorus and selenium. For girls in the same age range, there was also a high prevalence of inadequate vitamin B6 intake (50.4%) (Table 3). The mean sodium intake

**Table 1. General characteristics of adolescents from public schools.**

| Variables | | 95% CI[a] |
|---|---|---|
| Age, years (mean; SD[b]) | 11.8 (1.4) | 11.7–12.0 |
| Sex (n [%]) | | |
| Male | 223 (50.2) | 45.6–54.9 |
| Female | 221 (49.8) | 45.2–54.4 |
| Sexual maturation (n [%]) | | |
| Pre-pubescent | 164 (36.9) | 32.6–41.5 |
| Initial puberty | 113 (25.5) | 21.6–29.7 |
| Final puberty | 167 (37.6) | 33.2–42.2 |
| BMI[c] classification (n [%]) | | |
| Eutrophy | 332 (74.8) | 70.5–78.6 |
| Low weight | 18 (4.1) | 2.6–6.3 |
| Overweight | 62 (14.0) | 11.1–17.5 |
| Obese | 32 (7.2) | 5.2–10.0 |
| Maternal Education[d] (n [%]) | | |
| Not literate | 17 (4.6) | 2.9–7.1 |
| Primary school | 152 (40.5) | 35.7–45.6 |
| High school or higher education | 206 (54.9) | 49.9–59.9 |

[a]CI = confidence interval
[b]SD = standard deviation
[c]BMI = body mass index
[d]Values may not correspond to the total number of subjects in each group because of missing data

**Table 2. Contribution (%) of processed and ultra-processed food groups to the total energy intake in adolescents from public schools.**

| Food groups | Quartiles of percentage of energy from processed and ultra-processed foods | | | |
| --- | --- | --- | --- | --- |
| | Q1 | Q2 | Q3 | Q4 |
| | Mean (SD[a]) | Mean (SD) | Mean (SD) | Mean (SD) |
| *Processed* | | | | |
| French bread | 1.7 (0.4) | 4.0 (1.3) | 8.6 (1.4) | 16.4 (4.3) |
| Cheeses | 0.1 (0.0) | 0.2 (0.0) | 0.5 (0.2) | 4.0 (2.9) |
| Processed meats | 1.5 (0.4) | 3.0 (0.5) | 5.0 (0.7) | 9.9 (3.3) |
| Canned fruits and vegetables | 0.0 (0.0) | 0.0 (0.0) | 0.0 (0.0) | 0.2 (0.3) |
| **Total**[b] | 5.8 (1.7) | 10.4 (1.2) | 14.8 (1.5) | 20.6 (2.9) |
| *Ultra-processed* | | | | |
| Cakes, pies, and cookies | 7.5 (1.5) | 11.4 (1.0) | 14.7 (1.1) | 19.7 (2.5) |
| Fast food dishes | 0.4 (0.1) | 0.7 (0.1) | 1.3 (0.4) | 8.2 (3.7) |
| Sugar-sweetened beverages | 1.0 (0.2) | 1.8 (0.2) | 2.6 (0.3) | 3.9 (0.8) |
| Sliced breads | 0.9 (0.2) | 1.5 (0.2) | 5.9 (1.8) | 12.7 (3.4) |
| Bakery products | 0.8 (0.2) | 1.7 (0.3) | 3.1 (0.6) | 6.5 (2.5) |
| Snacks | 0.2 (0.1) | 0.4 (0.1) | 0.9 (0.3) | 6.8 (3.7) |
| Ultra-processed meats | 1.0 (0.3) | 1.9 (0.3) | 3.6 (0.5) | 6.3 (1.8) |
| Ready-to-eat and semi-ready-to-eat meals | 0.2 (0.0) | 0.3 (0.0) | 0.4 (0.0) | 3.6 (3.4) |
| Sweetened milk drinks | 0.2 (0.0) | 0.2 (0.0) | 0.3 (0.0) | 3.3 (1.5) |
| Other ultra-processed foods | 1.2 (0.4) | 2.5 (0.3) | 3.8 (0.4) | 6.0 (1.2) |
| **Total**[b] | 21.4 (4.9) | 31.5 (2.2) | 41.4 (3.9) | 61.5 (11.7) |

[a]SD = standard deviation

[b]Expressed as a percentage of total energy intake.

was high (3549.9 mg (948.1 mg) in boys and 3316.9 mg (838.7 mg) in girls), indicating that more than 93.5% of adolescents had sodium intake that exceeded the upper limit.

The percentage of energy consumed from processed foods was significantly associated with inadequate selenium intake, suggesting that adolescents who consumed the highest levels of processed foods were almost twice as likely to have inadequate selenium intake than subjects who consumed the lowest levels of processed foods ($p < 0.01$, PR = 1.97, 95% CI: 1.22–3.1). However, consumption of processed foods was associated with a lower probability of vitamin B1 inadequacy ($p = 0.04$, PR = 0.55, 95% CI: 0.31–0.98; Table 4). The percentage of energy consumed from ultra-processed foods was also significantly associated with a lower probability of vitamin B1 inadequacy ($p = 0.03$, PR = 0.49, 95% CI: 0.25–0.94). In addition, a lower probability of inadequate zinc intake ($p < 0.01$) was observed for subjects with the highest levels of energy consumption from ultra-processed foods ($p < 0.01$, PR = 0.43 95% CI 0.23–0.78), while no association was noted between zinc intake and processed food levels (Table 4). No association was found between the proportion of energy obtained from processed or ultra-processed foods and the prevalence of inadequate intake of other micronutrients (all $p > 0.05$; Table 4).

## Discussion

This study of students in public schools in an urban area in the city of Natal, northeastern Brazil, confirms an increase in the proportion of energy obtained from processed and ultra-processed foods in adolescent diets. This is consistent with the findings of other studies carried out in Brazil in the pediatric population [5, 10, 15]. The intake of processed foods showed significant associations with the prevalence of inadequacy of selenium intake. An increased

**Table 3. Daily nutritional recommendations, micronutrient intake, and prevalence of inadequacy of micronutrient intake (%IN[a]) by sex and age group in adolescents from public schools.**

| Micronutrients | Male[a] | | | | | | Female[b] | | | | | |
|---|---|---|---|---|---|---|---|---|---|---|---|---|
| | EAR[c]/UL[d] | Mean (SD[e]) | 10th | 50th | 90th | %IN[f] | EAR/UL | Mean (SD) | 10th | 50th | 90th | %IN |
| Vitamin A (μg) | | | | | | | | | | | | |
| 10 to 13 y | 445 | 885.3 (539.9) | 402.4 | 827.3 | 1275.3 | 20.9 | 420 | 822.5 (410.7) | 334.5 | 766.6 | 1308.4 | 16.3 |
| 14 to 18 y | 630 | 737.5 (248.7) | 293.1 | 787.2 | 1064.0 | 33.4 | 485 | 1077.9 (1008.0) | 410.9 | 731.9 | 2701.9 | 27.8 |
| Vitamin C (mg) | | | | | | | | | | | | |
| 10 to 13 y | 39 | 162.9 (130.8) | 55.6 | 118.0 | 338.7 | 17.1 | 39 | 161.5 (144.6) | 42.5 | 106.5 | 383.3 | 19.8 |
| 14 to 18 y | 63 | 166.4 (143.3) | 49.8 | 134.3 | 359.0 | 23.6 | 56 | 161.1 (180.1) | 34.7 | 100.8 | 523.1 | 28.1 |
| Vitamin B1 (mg) | | | | | | | | | | | | |
| 10 to 13 y | 0.7 | 1.1 (0.4) | 0.7 | 1.0 | 1.6 | 17.4 | 0.7 | 1.1 (0.4) | 0.6 | 1.0 | 1.5 | 21.2 |
| 14 to 18 y | 1.0 | 1.4 (0.8) | 0.8 | 1.3 | 1.8 | 30.2 | 0.9 | 0.9 (0.2) | 0.5 | 0.9 | 1.3 | 47.2 |
| Vitamin B2 (mg) | | | | | | | | | | | | |
| 10 to 13 y | 0.8 | 1.1 (0.4) | 0.7 | 1.1 | 1.7 | 21.5 | 0.8 | 1.1 (0.6) | 0.6 | 1.0 | 1.6 | 30.1 |
| 14 to 18 y | 1.1 | 1.3 (0.5) | 0.7 | 1.1 | 2.1 | 37.5 | 0.9 | 1.1 (0.4) | 0.6 | 0.9 | 1.6 | 33.4 |
| Vitamin B6 (mg) | | | | | | | | | | | | |
| 10 to 13 y | 0.8 | 0.9 (0.3) | 0.5 | 0.8 | 1.3 | 41.7 | 0.8 | 0.8 (0.3) | 0.5 | 0.8 | 1.2 | 50.4 |
| 14 to 18 y | 1.1 | 1.0 (0.3) | 0.6 | 0.9 | 1.5 | 62.9 | 1.0 | 0.8 (0.3) | 0.5 | 0.7 | 1.4 | 74.2 |
| Vitamin B12 (μg) | | | | | | | | | | | | |
| 10 to 13 y | 1.5 | 1.7 (1.9) | 0.4 | 1.2 | 3.5 | 46.4 | 1.5 | 1.5 (1.6) | 0.3 | 1.1 | 3.3 | 49.2 |
| 14 to 18 y | 2.0 | 1.3 (1.0) | 0.1 | 1.0 | 3.4 | 75.5 | 2.0 | 1.9 (1.6) | 0.3 | 1.7 | 5.8 | 51.6 |
| Vitamin D (μg) | | | | | | | | | | | | |
| 10 to 13 y | 10 | 0.1 (0.1) | 0.1 | 0.1 | 0.2 | 100 | 10 | 0.1 (0.1) | 0.1 | 0.1 | 0.3 | 100 |
| 14 to 18 y | 10 | 0.2 (0.1) | 0.1 | 0.2 | 0.3 | 100 | 10 | 0.1 (0.1) | 0.1 | 0.1 | 0.3 | 100 |
| Folate (μg) | | | | | | | | | | | | |
| 10 to 13 y | 250 | 89.8 (48.2) | 38.8 | 78.4 | 155.6 | 99.9 | 250 | 91.2 (57.2) | 33.4 | 77.0 | 163.4 | 99.7 |
| 14 to 18 y | 330 | 123.1 (87.5) | 22.7 | 109.7 | 291.1 | 99.1 | 330 | 86.4 (35.3) | 42.5 | 88.7 | 132.9 | 100 |
| Vitamin E (mg) | | | | | | | | | | | | |
| 10 to 13 y | 9.0 | 2.9 (1.9) | 1.0 | 2.4 | 5.6 | 99.9 | 9.0 | 3.1 (2.4) | 0.8 | 2.4 | 6.0 | 99.2 |
| 14 to 18 y | 12 | 3.3 (2.9) | 0.8 | 2.6 | 7.0 | 99.9 | 12 | 4.1 (3.7) | 1.1 | 2.7 | 8.4 | 98.4 |
| Sodium (mg)[g] | | | | | | | | | | | | |
| 10 to 13 y | 2200 | 3504.6 (919.6) | 2430.0 | 3369.8 | 4640.4 | – | 2200 | 3341.5 (850.4) | 2333.9 | 3310.4 | 4529.0 | – |
| 14 to 18 y | 2300 | 4067.1 (1132.5) | 2506.1 | 3972.1 | 5779.0 | – | 2300 | 3039.4 (648.9) | 2199.8 | 2927.2 | 4193.7 | – |
| Calcium (mg) | | | | | | | | | | | | |
| 10 to 13 y | 1100 | 398.8 (140.1) | 249.4 | 367.2 | 620.1 | 100 | 1100 | 380.4 (128.6) | 221.0 | 353.0 | 561.0 | 100 |
| 14 to 18 y | 1100 | 439.3 (117.2) | 292.8 | 436.9 | 563.8 | 100 | 1100 | 371.3 (125.6) | 188.9 | 385.8 | 532.7 | 100 |
| Zinc (mg) | | | | | | | | | | | | |
| 10 to 13 y | 7.0 | 10.5 (3.4) | 6.8 | 10.0 | 14.5 | 15.2 | 7.0 | 9.4 (2.7) | 5.9 | 9.3 | 12.8 | 18.7 |
| 14 to 18 y | 8.5 | 11.4 (2.8) | 7.6 | 11.3 | 14.9 | 15.2 | 7.3 | 8.5 (1.4) | 6.6 | 8.7 | 10.1 | 19.2 |
| Phosphorus (mg) | | | | | | | | | | | | |
| 10 to 13 y | 1055 | 876.4 (198.0) | 647.6 | 845.0 | 1165.1 | 81.6 | 1055 | 829.0 (189.5) | 600.8 | 827.1 | 1072.7 | 88.3 |
| 14 to 18 y | 1055 | 959.4 (180.8) | 768.9 | 919.0 | 1293.2 | 70.2 | 1055 | 811.9 (175.9) | 561.0 | 835.6 | 1053.3 | 91.6 |
| Iron (mg) | | | | | | | | | | | | |
| 10 to 13 y | 5.9 | 11.9 (3.2) | 8.1 | 11.5 | 16.3 | 1.3 | 5.7 | 11.1 (3.0) | 7.4 | 10.9 | 15.1 | 3.6 |
| 14 to 18 y | 7.7 | 13.9 (4.8) | 7.4 | 14.6 | 19.8 | 11.1 | 7.9 | 10.1 (2.8) | 7.4 | 9.2 | 14.0 | 29.2 |
| Selenium (μg) | | | | | | | | | | | | |
| 10 to 13 y | 35 | 26.2 (9.9) | 14.3 | 25.2 | 40.3 | 81.3 | 35 | 25.3 (9.9) | 13.6 | 24.5 | 37.0 | 83.6 |

*(Continued)*

**Table 3.** (Continued)

| Micronutrients | Male[a] | | | | | | Female[b] | | | | | |
|---|---|---|---|---|---|---|---|---|---|---|---|---|
| | EAR[c]/UL[d] | Mean (SD[e]) | 10th | 50th | 90th | %IN[f] | EAR/UL | Mean (SD) | 10th | 50th | 90th | %IN |
| 14 to 18 y | 45 | 30.9 (15.1) | 12.4 | 28.4 | 61.4 | 82.4 | 45 | 24.3 (7.4) | 11.5 | 24.1 | 33.5 | 99.7 |

[a]male 10 to 13 years (n = 205); 14 to 18 years (n = 18)

[b]female 10 to 13 years (n = 203); 14 to 18 years (n = 18)

[c]EAR = estimated average requirement

[d]UL = tolerable upper intake level

[e]SD = standard deviation

[f]IN = inadequate

[g]sodium intake was analyzed using UL values.

proportion of energy from processed and ultra-processed foods was also associated with a lower probability of vitamin B1 inadequacy. In addition, energy consumption from ultra-processed foods was associated with a lower probability of inadequate zinc intake.

The Brazilian government National Adolescent School-based Health Survey (PeNSE 2015) reinforces that this new food behavior affects all of Brazil's socioeconomic levels and regions, considering that 40% of schoolchildren reported daily consumption of at least one group of ultra-processed foods [19]. Accordingly, strategies to promote healthy eating and decrease sedentary behavior, as well as regulation of advertising for ultra-processed foods, are necessary to prevent unhealthy lifestyles from persisting into adulthood [3].

There were low percentages of energy intake from ready-to-eat and semi-ready-to-eat meals and sweetened milk drinks from the ultra-processed foods group. This finding could be attributed to the fact that most families living in northeastern Brazil are characterized as low-income families [41, 42], as an increase in the intake of ready-to-eat and semi-ready-to-eat meals and sweetened milk drinks has been associated with an increase in family income per capita [42].

Previous work points out that diets based on traditional Brazilian foods, characterized by high intake of beans and rice, are more frequent among adolescents [43]. The Pure Traditional Food System pattern, which consists of beans and a mixture of characteristic ingredients (e.g., onions, peppers, and tomatoes) was predominant in this study population of adolescents from northeastern Brazil [22]. Apart from beef, the foods comprising this pattern are of low cost, and few food items were associated with increased micronutrient intake, suggesting monotony and low nutritional quality in the diet of the less privileged socioeconomic classes [43].

Adolescents' intake of these lower cost foods with low nutritional quality may be associated with a higher prevalence of vitamin D, calcium, and phosphorus intake inadequacy, with implications for bone health in this stage of life [44, 45]. The high prevalence of inadequate vitamin D intake observed in our study indicates an alarming situation, since one of the main actions of vitamin D is the regulation of calcium and phosphorus metabolism, controlling the processes of intestinal absorption and renal reabsorption of these ions [45].

The high prevalence of inadequate calcium intake observed in our study may be partly explained by the persistently low consumption of dairy products among adolescents. Other Brazilian studies have also shown low milk consumption and increased soft drinks or sugary drinks consumption [46, 47]. Similarly, Mexican adolescents aged 12–19 years showed high calcium inadequacy, reaching 88.1% in females and 71.8% in males [48]. During adolescence, there is accelerated growth, with greater anthropometric and body composition changes,

**Table 4. Models of the unadjusted and adjusted association between proportion of energy from processed and ultra-processed foods and the prevalence of inadequate micronutrient intake in adolescents from public schools.**

| Parameter[a] | Unadjusted Model | | | Adjusted Model[b] | | | |
|---|---|---|---|---|---|---|---|
| | PR[c] | 95% CI[d] | P-value | $\chi^2$ of Wald | P-value | PR | 95% CI |
| Processed foods[e] | | | | | | | |
| 1st quartile | | | | 4.09 | 0.04 | 12.96 | (1.08–155.27) |
| 2nd quartile | | | | 1.23 | 0.26 | 4.06 | (0.34–48.38) |
| 3rd quartile | | | | 0.04 | 0.84 | 1.28 | (0.10–15.29) |
| 4rd quartile | 1 | | | | | | |
| Vitamin A | | | | | | | |
| inadequate | 1.39 | (0.86–2.24) | 0.17 | 1.14 | 0.28 | 1.31 | (0.79–2.17) |
| adequate | 1 | | | | | 1 | |
| Vitamin B1 | | | | | | | |
| inadequate | 0.58 | (0.36–0.94) | 0.02 | 4.06 | 0.04 | 0.55 | (0.31–0.98) |
| adequate | 1 | | | | | 1 | |
| Vitamin B2 | | | | | | | |
| inadequate | 0.73 | (0.50–1.07) | 0.10 | 0.43 | 0.51 | 0.84 | (0.51–1.38) |
| adequate | 1 | | | | | 1 | |
| Vitamin B6 | | | | | | | |
| inadequate | 0.79 | (0.56–1.10) | 0.17 | 1.53 | 0.21 | 0.76 | (0.50–1.16) |
| adequate | 1 | | | | | 1 | |
| Vitamin B12 | | | | | | | |
| inadequate | 1.12 | (0.86–1.74) | 0.25 | 0.84 | 0.35 | 1.19 | (0.81–1.73) |
| adequate | 1 | | | | | 1 | |
| Folate | | | | | | | |
| inadequate | 2.87 | (0.45–18.32) | 0.26 | 0.45 | 0.50 | 1.88 | (0.29–11.87) |
| adequate | 1 | | | | | 1 | |
| Phosphorus | | | | | | | |
| inadequate | 1.70 | (1.04–2.78) | 0.03 | 2.53 | 0.11 | 1.56 | (0.90–2.71) |
| adequate | 1 | | | | | 1 | |
| Selenium | | | | | | | |
| inadequate | 2.05 | (1.31–3.21) | <0.01 | 7.69 | <0.01 | 1.97 | (1.22–3.17) |
| adequate | 1 | | | | | 1 | |
| Ultra-processed foods[f] | | | | | | | |
| 1st quartile | | | | 4.53 | 0.033 | 0.149 | (0.02–0.85) |
| 2nd quartile | | | | 12.01 | 0.001 | 0.044 | (0.008–0.25) |
| 3rd quartile | | | | 22.12 | 0.001 | 0.014 | (0.002–0.08) |
| 4rd quartile | 1 | | | | | | |
| Vitamin C | | | | | | | |
| inadequate | 0.37 | (0.18–0.75) | <0.001 | 2.40 | 0.12 | 0.54 | (0.25–1.17) |
| adequate | 1 | | | | | 1 | |
| Vitamin B1 | | | | | | | |
| inadequate | 0.33 | (0.19–0.57) | <0.0001 | 4.56 | 0.03 | 0.49 | (0.25–0.94) |
| adequate | 1 | | | | | 1 | |
| Vitamin B2 | | | | | | | |
| inadequate | 0.69 | (0.47–1.02) | 0.06 | 2.47 | 0.11 | 1.50 | (0.90–2.50) |
| adequate | 1 | | | | | 1 | |
| Vitamin B6 | | | | | | | |
| inadequate | 0.54 | (0.38–0.75) | <0.001 | 3.04 | 0.08 | 0.68 | (0.45–1.04) |

(*Continued*)

**Table 4.** (Continued)

| Parameter[a] | Unadjusted Model | | | Adjusted Model[b] | | | |
|---|---|---|---|---|---|---|---|
| | PR[c] | 95% CI[d] | P-value | χ² of Wald | P-value | PR | 95% CI |
| adequate | 1 | | | | | 1 | |
| Vitamin E | | | | | | | |
| inadequate | 0.45 | (0.17–1.20) | 0.11 | 2.51 | 0.11 | 0.44 | (0.16–1.21) |
| adequate | 1 | | | | | 1 | |
| Sodium | | | | | | | |
| inadequate | 0.27 | (0.12–0.59) | 0.001 | 0.60 | 0.43 | 0.69 | (0.28–1.73) |
| adequate | 1 | | | | | 1 | |
| Zinc | | | | | | | |
| inadequate | 0.29 | (0.17–0.48) | <0.001 | 7.60 | <0.01 | 0.43 | (0.23–0.78) |
| adequate | 1 | | | | | 1 | |
| Iron | | | | | | | |
| inadequate | 0.10 | (0.02–0.49) | <0.01 | 0.73 | 0.39 | 0.47 | (0.08–2.65) |
| adequate | 1 | | | | | 1 | |

[a]Q1 vs Q4 comparison for each micronutrient

[b]The final model was adjusted with all the variables presented in the table.

[c]PR = prevalence ratio

[d]CI = confidence interval

[e]Vitamins D, C, and E, and the minerals calcium, zinc, sodium, and iron were excluded from the adjusted model since they were not significant in the unadjusted analysis (p > 0.30).

[f]Vitamins A, B12, D and E, folate, and the minerals calcium, phosphorous, and selenium were excluded from the adjusted model since they were not significant in the unadjusted analysis (p > 0.30).

which require an adequate supply of calcium, vitamin D, and phosphorus. These nutrients are essential in guaranteeing the achievement of growth and development [49].

The high prevalence of inadequate vitamin E and selenium intake was similar to those of the ERICA findings, reflecting the choice of ultra-processed foods with lower micronutrient content [18]. These micronutrients are recognized for their antioxidant roles in the body, and act synergistically to maintain glutathione peroxidase capacity. Importantly, an imbalance in these nutrients can favor oxidative stress, causing dysregulation in signaling and/or cell damage [50].

Food fortification with folate has been implemented in several countries. In Brazil, fortification of wheat and maize flours has been implemented since 2004, with the addition of 150 mg of folic acid and 4.2 mg of iron per 100 g of flour [51]. In this study, although adolescents presented with a high intake of processed and ultra-processed foods made with flour, a high prevalence of folate inadequacy was observed. Fiorentino et al. [52] also confirmed a high prevalence of folate intake inadequacy among Senegalese children and adolescents. In our study, this fact could be attributed to a lack of information on folate in Brazilian food composition tables [32], including fortified foods. Therefore, the prevalence of folate intake inadequacy has been overestimated.

The prevalence of iron intake inadequacy in girls is notable because, during menarche, dietary iron requirements are increased [53]. However, as observed with folate intake, the prevalence of iron inadequacy, while low, may be overestimated, considering the possible failures of Brazilian food composition tables [32].

Our results regarding excessive sodium intake confirm previous findings obtained in Brazil [18, 54] and Europe [8]. Although the adjusted model did not reach the level of significance to

suggest an association between processed and ultra-processed food consumption and sodium intake, the high sodium intake observed among the adolescents in this study may be related to high intake of these types of foods. This is worrying because high sodium intake among adolescents may be associated with risk of cardiometabolic disease [55].

The most significant finding we observed pertained to the association between processed foods consumption and selenium intake inadequacy. Louzada et al. [56] also observed a relationship between ultra-processed food intake and low consumption of selenium in Brazilians aged over 10 years. Different authors have pointed out the difficulties in assessing selenium content in food because it depends on climatic conditions, cultivation and breeding methods. The concentration of this element in foods also varies depending on the species, plant part, concentration of selenium in the soil, and the ability of plants to accumulate this element [57]. In our study, we observed low intake of foods that are significant sources of selenium, such as fish, whole grains, and Brazil nuts. A previous study has shown that selenium inadequacy negatively affects the thyroid metabolism of iodine-replete children and may present a substantial public health concern. The authors reinforce the importance of selenium status for normal thyroid function, as well as others roles of this element in biological systems [58].

An unexpected finding in our study was the associations between the consumption of processed and ultra-processed food and lower probability of inadequate vitamin B1. In addition, a higher consumption of ultra-processed foods was also significantly associated with a lower probability of inadequate zinc intake. It is known that in the process of producing white flour, much of vitamin content is lost [59]. In the present study, a higher intake of foods produced with white flour, such as cakes, pies, and biscuits, as well as breads, was observed as the most commonly consumed food group among subjects from both groups. However, a possible explanation for this association is the fact that the loss of vitamins during food processing has led the industry to add vitamins and minerals to various foodstuffs, with the goal of reducing nutritional deficiencies in the population [60]. This fortification may explain the observed association between high consumption of processed foods and lower probability of inadequacy of vitamin B1 and zinc.

A previous study conducted with a group of Brazilian children aged 2–3 years has indicated a low prevalence of iron, vitamin C, vitamin A, calcium, and folate inadequacy. None of these children had intakes less than the EAR value for zinc. However, 4·0% of children exceeded the UL for vitamin A, 3.1% for zinc, 1.1% for folic acid, and 0.2% for iron. The authors emphasize that the consumption of ultra-processed foods, usually fortified foods, contributed to micronutrient supply, but may increase the risk of excessive micronutrient intake [61]. Authors point out that enriched or fortified foods in the US are an important source of micronutrients. According to this line of reasoning, variation in ultra-processed food consumption does not necessarily result in nutritional imbalances [62].

Previous studies have addressed the controversy over the association between consumption of ultra-processed foods and the proportion of individuals whose intakes are below the micronutrient requirements, as well as the difficulty in setting adequate and safe micronutrient levels in foods without increasing the risk of consuming amounts above the UL [63]. The bioavailability of micronutrients depends on food composition and possibly on its encapsulation method in some specific formulations. Furthermore, ultra-processed foods do not provide the same health benefits as natural foods [4]. Beyond the addition of synthetic micronutrients that can exceed the UL, ultra-processed foods are energy-dense, with high levels of fat, sugar, and/or sodium, contributing to the increased risk of obesity and noncommunicable diseases [64].

Our study has some limitations. We faced difficulties in assessing food consumption due to the incompleteness of the food composition tables of available foods. However, statistical techniques were used to minimize the probability of the occurrence of random errors. Another

limitation is the number of days taken to evaluate the habitual intake of nutrients, since this number may vary between nutrients. Therefore, evaluation of the habitual intake of some of the micronutrients may be imprecise.

## Conclusions

In conclusion, an increase in the proportion of energy obtained from processed and ultra-processed foods by adolescents from public schools, was associated with an increase in the prevalence of inadequate selenium intake, which could lead to short-term and long-term negative health consequences. However, energy consumption from processed and ultra-processed foods was associated with a lower probability of inadequate vitamin B1 intake. In addition, a lower probability of inadequate zinc intake was observed only for energy consumption from ultra-processed foods, possibly attributed to food fortification. Considering the risk of micronutrient intake exceeding the UL, and the fact that ultra-processed foods are energy-dense, with high levels of fat, sugar, and/or sodium, we believe that it is crucial to stress the importance of limiting personal consumption of processed and ultra-processed foods, especially among the pediatric population. The position of the Academy of Nutrition and Dietetics highlights that early care and education programs should recommended, from an early age, a high standard of nutritional quality for foods and beverages served. Efforts to promote healthy eating should be a national priority and include state and local policy changes [65]. Accordingly, the results of this study can help provide evidence for formulating global public policies that improve health in adolescence and prevent complications in adulthood.

## Supporting information

**S1 Table. Classification of foods and ingredients according to their industrial processing characteristics.** [a]Adapted from Louzada et al.[37].
(DOCX)

## Acknowledgments

We thank undergraduate students of the Nutrition Undergraduate at the Federal University of Rio Grande do Norte who contributed substantially to the assembly of the food consumption bank and scientific discussions.

## Author Contributions

**Conceptualization:** Raphaela Cecília Thé Maia de Arruda Falcão, Clélia de Oliveira Lyra, Lucia Fátima Campos Pedrosa, Severina Carla Vieira Cunha Lima, Karine Cavalcanti Maurício Sena-Evangelista.

**Data curation:** Raphaela Cecília Thé Maia de Arruda Falcão, Clélia de Oliveira Lyra, Célia Márcia Medeiros de Morais, Lucia Fátima Campos Pedrosa, Severina Carla Vieira Cunha Lima, Karine Cavalcanti Maurício Sena-Evangelista.

**Formal analysis:** Raphaela Cecília Thé Maia de Arruda Falcão, Clélia de Oliveira Lyra, Célia Márcia Medeiros de Morais, Severina Carla Vieira Cunha Lima, Karine Cavalcanti Maurício Sena-Evangelista.

**Funding acquisition:** Lucia Fátima Campos Pedrosa, Karine Cavalcanti Maurício Sena-Evangelista.

**Investigation:** Raphaela Cecília Thé Maia de Arruda Falcão, Liana Galvão Bacurau Pinheiro, Severina Carla Vieira Cunha Lima, Karine Cavalcanti Maurício Sena-Evangelista.

**Methodology:** Raphaela Cecília Thé Maia de Arruda Falcão, Clélia de Oliveira Lyra, Célia Márcia Medeiros de Morais, Liana Galvão Bacurau Pinheiro, Lucia Fátima Campos Pedrosa, Severina Carla Vieira Cunha Lima, Karine Cavalcanti Maurício Sena-Evangelista.

**Project administration:** Karine Cavalcanti Maurício Sena-Evangelista.

**Supervision:** Lucia Fátima Campos Pedrosa, Severina Carla Vieira Cunha Lima, Karine Cavalcanti Maurício Sena-Evangelista.

**Validation:** Karine Cavalcanti Maurício Sena-Evangelista.

**Visualization:** Karine Cavalcanti Maurício Sena-Evangelista.

**Writing – original draft:** Raphaela Cecília Thé Maia de Arruda Falcão, Clélia de Oliveira Lyra, Célia Márcia Medeiros de Morais, Liana Galvão Bacurau Pinheiro, Lucia Fátima Campos Pedrosa, Severina Carla Vieira Cunha Lima, Karine Cavalcanti Maurício Sena-Evangelista.

**Writing – review & editing:** Raphaela Cecília Thé Maia de Arruda Falcão, Clélia de Oliveira Lyra, Célia Márcia Medeiros de Morais, Liana Galvão Bacurau Pinheiro, Lucia Fátima Campos Pedrosa, Severina Carla Vieira Cunha Lima, Karine Cavalcanti Maurício Sena-Evangelista.

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
