## [Decision Letter · Decision Letter 0]

27 Aug 2019

PONE-D-19-18574

Processed and ultra-processed foods are associated with high prevalence of inadequate selenium intake and low prevalence of vitamin B1 and zinc inadequacy in adolescents from public schools

PLOS ONE

Dear Dr. Sena-Evangelista,

Thank you for submitting your manuscript to PLOS ONE. After careful consideration, we feel that it has merit but does not fully meet PLOS ONE’s publication criteria as it currently stands. Therefore, we invite you to submit a revised version of the manuscript that addresses the points raised during the review process.

The manuscript addressed an important public health nutrition problem (consumption of processed and ultra-processed foods) in a vulnerable (adolescent) population and present the association between consumption of processed and ultra-processed foods and intake of selected micronutrients. The manuscript is also well written and properly organized. However, as stated below, I have serious concerns on the approach of analysis employed.

1. Line 242-3: “……the energy percentage from processed and ultra-processed foods and the prevalence of inadequate micronutrient intake was developed using generalized linear models with an ordinal logistic distribution”. It is not clear which one of the two is the dependent variable of the analysis (prevalence of inadequate micronutrient intake or percentages of energy obtained from processed and ultra-processed foods). As long as the purpose of the study is to see the association between consumption of processed and ultra-processed foods (kind of exposure) with the prevalence of inadequate micronutrient intake (kind of outcome); micronutrient intake should be the outcome/dependent variable. And what you should have used is binary not ordinary logistic regression analysis.

2. I really don’t see the purpose of adjusting one micronutrient nutrient intake for the other. Do you really expect confounding among them? What I propose is multiple bivariable models for each nutrient, by which the dependent variable the nutrient intake and the independent variable is the percentages of energy obtained from processed and ultra-processed foods.

We would appreciate receiving your revised manuscript by Oct 11 2019 11:59PM. To enhance the reproducibility of your results, we recommend that if applicable you deposit your laboratory protocols in protocols.io, where a protocol can be assigned its own identifier (DOI) such that it can be cited independently in the future. For instructions see: http://journals.plos.org/plosone/s/submission-guidelines#loc-laboratory-protocols

We look forward to receiving your revised manuscript.

Kind regards,

Samson Gebremedhin Gebreselassie, PhD

Academic Editor

PLOS ONE

Journal Requirements:

- Kieliszek, Marek, and Stanisław Błażejak. "Current knowledge on the importance of selenium in food for living organisms: a review." Molecules 21.5 (2016): 609.

- Gashu, Dawd, et al. "Selenium inadequacy hampers thyroid response of young children after iodine repletion." Journal of Trace Elements in Medicine and Biology 50 (2018): 291-295.

- Sangalli, Caroline Nicola, Fernanda Rauber, and Márcia Regina Vitolo. "Low prevalence of inadequate micronutrient intake in young children in the south of Brazil: a new perspective." British Journal of Nutrition 116.5 (2016): 890-896.

 The text that needs to be addressed involves some sentences of the Discussion.

In your revision ensure you cite all your sources (including your own works), and quote or rephrase any duplicated text outside the methods section. Further consideration is dependent on these concerns being addressed.

Additional Editor Comments:

Title: Please indicate the setting in the title

Abstract

• In the abstract please concisely indicate how foods were classified into processed and ultra-processed foods. You may simple indicate the “NOVA” classification system was used.

• Background: Line 69-73; can you provide some common examples for processed and ultra-processed foods so that readers can easily understand the differences between the two?

Methods:

• line 141 and 150: it is not clear why the study considered primary school students as source population while the age group of interest is 10-19 years of age.

• Line 153-154: ample size is calculated based on the assumption that the outcome of interested is altered lipid profiles in each district and this is not directly related with the current manuscript. How do you assure the adequacy of the sample size/power to address the objective of this specific manuscript?

• Line 189-90: the sentence “Recall surveys were performed at an interval of 30–45 days, according to the recommendation of Thompson and Byers” is not clear. Please provide additional description of this approach.

• Line 247: “Variables with p-values ≤ 0.30…….” can you please provide the list of variables considered for the bivariable analysis? As I stated above, I really question the importance of multivariable analysis here.

• Line 245-7: As I commented above, please also clearly indicate the dependent and independent variables of the model.

Results

• Please provide the operational definitions employed to classify the sexual maturation of adolescents.

• Line 303: “high prevalence of pyridoxine intake” confusing, high or low intake?

• Table 3: can you please explain why you limit yourself to the 15 micronutrients provided in the table?

• 97% more likely > almost two times more likely.

• Why it was not possible to present table 4 and 5 together in one table?

• Table 4 and 5: Please clearly describe the dependent and independent variables, I also propose for removing the multivariable analysis.

Table 4 and 5: as long as p values are there I don’t see the purpose of having the Wald statistic there.

Others

• Multivariate > Multivariable; bivariate > bivariable

Reviewers' comments:

Reviewer's Responses to Questions

**Comments to the Author**

1. Is the manuscript technically sound, and do the data support the conclusions?

Reviewer #1: Partly

Reviewer #2: Partly

Reviewer #3: Yes

2. Has the statistical analysis been performed appropriately and rigorously? 

Reviewer #1: Yes

Reviewer #2: Yes

Reviewer #3: Yes

3. Have the authors made all data underlying the findings in their manuscript fully available?

Reviewer #1: Yes

Reviewer #2: No

Reviewer #3: Yes

4. Is the manuscript presented in an intelligible fashion and written in standard English?

Reviewer #1: No

Reviewer #2: Yes

Reviewer #3: Yes

5. Review Comments to the Author

Reviewer #1: Processed and ultra-processed foods are associated with high prevalence of inadequate

selenium intake and low prevalence of vitamin B1 and zinc inadequacy in adolescents from

public schools.

The study addresses an interesting and relevant subject showing the association of consumption of processed and ultra-processed foods with micronutrient inadequacy in adolescents. However, the MS lacks logical presentation of information hence is difficult to understand.

The introduction doesn’t contain the relevant background information to guide readers to the rest of the manuscript.

The justification or motivation to do this research is not well formulated.

The language not clear.

The outcome of this study mainly depends on the quality of the food composition table used to convert into nutrient intake of the present study subjects. However, this may not work for some nutrients such as selenium. This is because; selenium concentration in foods is mainly determined by the amount of the element in the soil which is quite variable in short distances. Thus, data from food composition is not reliable.

Parts of the methodology lack clarity and are not detailed.

The authors in general tried to show the effect of level of processing on micronutrient intake inadequacy that consumption of processed foods was associated with lower intake of selenium but not Zn and B1. Processed foods in general are poor in micronutrient content. However, it is failed to identify whether foods in the present study were enriched with Zn and vitamin B1 concentration.

For example,

Line 48-50, Energy consumption from processed foods was associated with higher prevalence of inadequate selenium intake (p < 0.01) but lower prevalence of inadequate vitamin B1 intake (p = 0.04).

Line 50-54, Energy consumption from ultra-processed foods was associated with lower prevalence of inadequate zinc and vitamin B1 intake (p < 0.01 and p =0.03, respectively). An increase in the proportion of energy obtained from processed and ultra-processed foods may reflect higher prevalence of inadequate selenium intake and lower prevalence of vitamin B1 and zinc inadequacy.

Line 74-75. It is not always true to claim that sedentary behaviors among adolescents lead to greater consumption of packed or ready to eat foods.

Methodology

Line 136-140: Ethical consideration/approval has to be presented separately

Line 141-148: The inclusion and Inclusion criteria may work for the main study “Risk factors for cardiovascular disease among adolescent beneficiaries of the Brazilian government National School Lunch Program (PNAE)” but not for this specific MS.

The sample size was determined considering altered lipid profile as an outcome which is not consistent with the present study outcome.

Weight and height measurement procedure is not detailed

Line 179: ‘The morning was chosen to more easily enable methodological procedures’. This is not clear. Does this mean ‘anthropometric measurement was done during the morning? Why?

Line 182-185: It is not clear the importance of assessing pubertal stage given the age of study participants was known for this particular study.

Reviewer #2: 1) Abstract

2) Introduction - it would be good to also have some systematic review to be cited to reiterate how the unhealthy dietary intake from low middle income countries/ Asian as well to show this is a global burden.

3) - It is noted in methods the age of interest is 10-19. However, based on the results it appears only mean age 11.8 years. Why the age group seems to be specific (10-13)?

- Explain why just use 24 hours diet recall and repeated twice. Is it the most suitable for adolescents studies? Since your interest is micronutrient and ultra processed food, which method of dietary intake should be more robust to be used for this adolescents group? The inadequate number of days for this might influence why you discovered such results.

- what was the study response rate?

How reliable is the reported Tanner staging? Any confirmation by the paediatrician?

Results: The age group for 14 and above only 18? why is it so low? If this group is excluded, how different the results will be in table 3.

Figure - Out of 4457 subjects only 700+ enrolled. Why?

Since the two-stage random stratified sampling is used, why complex sample analysis not conducted?

It appears highlighted micronutrient intake was low but do you have biomarkers to associate with?

Discussion: If ultra processed will lead to reduce certain micronutrient content, will enrichment with that vitamins in the food minimise the problem of suboptimal nutrient?

Research has shown the association ultra-processed foods that contain added sugars, excess sodium, and unhealthful fats possibly lead to poor health outcome and is this pattern the same with yours?

In the conclusion - "stress the importance of limiting personal consumption of processed and ultra-processed", how and where at school, home or institution and what is the allowed value? If there is any? What would be the most suitable take home message?

Reviewer #3: Abstract

-Numerous sentences, such as the one highlighted below, starting with 'this':

"This was a cross-sectional study.”

-The term 'processed and ultra-processed foods’ is being used several times. It is very repetitive.

-Insert the acronym for 'estimated average requirement

Introduction

-The information stated in lines 96 and 97 has already been said in the introduction’s section elsewhere.

-The paper discusses that some results found might be due to differences of eating habits or economic status of northeastern Brazil, in comparison to the rest of the country, but it does not mention any of this in the introduction. The authors need to expand on this.

Methods

-Were rural schools included? The access to milk and dairy products could be different compared to the population from the urban ones.

-In Fig 1, why are there 700 adolescents enrolled when it is mentioned above only 11 were transferred from school?

-What about those participants who were taking dietary supplements? Was this type of information collected or evaluated?

-Was the apportioning and quantification of food reviewed by the 24-hour dietary recalls' interviewers ?

Discussion

-"We observed low percentages of energy intake from ready-to-eat and semi-ready-to- eat meals and sweetened milk drinks from the ultra-processed foods group.” I would prefer the passive voice.

-"We observed low percentages of energy intake from ready-to-eat and semi-ready-to- eat meals and sweetened milk drinks from the ultra-processed foods group. This finding could be attributed to the fact that most families living in northeastern Brazil are characterized as low-income families [39, 40], as an increase in the intake of ready-to-eat and semi-ready-to- eat meals and sweetened milk drinks has been associated with an increase in family income per capita [40].” Is it really different from "cakes, pies, and cookies"?

-“In our study, we observed low intake of foods that are significant sources of selenium, such as fish, whole grains, and Brazil nuts.” Can this information be drawn from table 2?

-“Brazilian children aged 2–3 years typically have a low prevalence of iron, vitamin C, vitamin A, calcium, and folate inadequacy. None of these children had intakes less than the EAR value for zinc. Fortified foods contributed to micronutrient supply. However, 4·0% of children exceeded the UL for vitamin A, 3.1% for zinc, 1.1% for folic acid, and 0.2% for iron. These results suggest a low prevalence of inadequate micronutrient intake among children, with the implication that this group could be at risk of excessive micronutrient intake provided by ultra-processed foods [59].” Isn't this paragraph deviating from the population targeted at this paper?

-According to results presented, processed and ultra-processed foods are associated with a low prevalence of vitamin B1 and zinc inadequacy. The authors need to expand on this.

-What are the limitations of the 24-hour dietary recall?

-The results need to be further compared with the literature.

6. PLOS authors have the option to publish the peer review history of their article (what does this mean?). If published, this will include your full peer review and any attached files.

Reviewer #1: No

Reviewer #2: No

Reviewer #3: No

---

## [Author Response · Author response to Decision Letter 0]

14 Oct 2019

Editor

Thank you for submitting your manuscript to PLOS ONE. After careful consideration, we feel that it has merit but does not fully meet PLOS ONE’s publication criteria as it currently stands. Therefore, we invite you to submit a revised version of the manuscript that addresses the points raised during the review process.

The manuscript addressed an important public health nutrition problem (consumption of processed and ultra-processed foods) in a vulnerable (adolescent) population and present the association between consumption of processed and ultra-processed foods and intake of selected micronutrients. The manuscript is also well written and properly organized. However, as stated below, I have serious concerns on the approach of analysis employed.

Line 242-3: “……the energy percentage from processed and ultra-processed foods and the prevalence of inadequate micronutrient intake was developed using generalized linear models with an ordinal logistic distribution”. It is not clear which one of the two is the dependent variable of the analysis (prevalence of inadequate micronutrient intake or percentages of energy obtained from processed and ultra-processed foods). As long as the purpose of the study is to see the association between consumption of processed and ultra-processed foods (kind of exposure) with the prevalence of inadequate micronutrient intake (kind of outcome); micronutrient intake should be the outcome/dependent variable. And what you should have used is binary not ordinary logistic regression analysis.

Answer: We would like to thank the editor for this comment. The purpose of the study is to evaluate the association between consumption of processed and ultra-processed foods with the prevalence of inadequate micronutrient intake. Considering the cross-sectional design of this study, one of the disadvantages is that the researcher does not know temporally what happens first, in this case, inadequacy of nutrients or consumption of processed and ultra-processed foods. We examine only the association and not the cause-effect relationship. Even if there is the assumption of this cause-effect relationship, there is no way to prove which occurred first. This methodological choice was made considering that consumption possibly does not interfere alone with only one micronutrient, and this effect needs to be controlled. Thus, with a cross-sectional design, the variables are placed interchangeably on different sides of the equation.

In addition, the bivariate or crude analysis was performed and it is as shown in Table 4 and specified as “Unadjusted Model”. Initially, unadjusted models were constructed in a bivariate analysis to identify associations between the prevalence of inadequate micronutrient intake and food energy percentage of processed and ultra-processed foods. Cross-sectional studies with binary outcomes analyzed by logistic regression are frequent in the epidemiological literature1.

1Barros AJ, Hirakata VN. Alternatives for logistic regression in cross-sectional studies: an empirical comparison of models that directly estimate the prevalence ratio. BMC Med Res Methodol. 2003 Oct 20;3:21.

2. I really don’t see the purpose of adjusting one micronutrient nutrient intake for the other. Do you really expect confounding among them? What I propose is multiple bivariable models for each nutrient, by which the dependent variable the nutrient intake and the independent variable is the percentages of energy obtained from processed and ultra-processed foods.

Answer: We are thankful to the editor for this comment. Multiple bivariate analyses cause a statistical problem called “type I error inflation” according to biomedical and biopsychosocial theoretical models, there is an interrelationship between health conditions. Thus, regarding food intake, the authors considered that it is necessary to control the covariance of inadequate micronutrient intake, as inadequate vitamin A intake may be possibly related to inadequate vitamin C intake, for example. 

We noticed you have some minor occurrence of overlapping text with the following previous publication(s), which needs to be addressed:

- Kieliszek, Marek, and Stanisław Błażejak. "Current knowledge on the importance of selenium in food for living organisms: a review." Molecules 21.5 (2016): 609.

- Gashu, Dawd, et al. "Selenium inadequacy hampers thyroid response of young children after iodine repletion." Journal of Trace Elements in Medicine and Biology 50 (2018): 291-295.

- Sangalli, Caroline Nicola, Fernanda Rauber, and Márcia Regina Vitolo. "Low prevalence of inadequate micronutrient intake in young children in the south of Brazil: a new perspective." British Journal of Nutrition 116.5 (2016): 890-896.

The text that needs to be addressed involves some sentences of the Discussion.

In your revision ensure you cite all your sources (including your own works), and quote or rephrase any duplicated text outside the methods section. Further consideration is dependent on these concerns being addressed.

Answer: We are thankful to the editor for this comment. The overlapping text has already been rewritten in the Discussion section (Page 26, lines 451-455; lines 458-460; Page 26 and 27, lines 473-480).

Additional Editor Comments:

Title: Please indicate the setting in the title

Answer: We appreciate the editor’s comment. We have added this information in the “Title section “…in an urban area of northeastern Brazil”. 

Abstract

In the abstract please concisely indicate how foods were classified into processed and ultra-processed foods. You may simple indicate the “NOVA” classification system was used.

Answer: We are thankful to the editor for this comment. The information has been added in the “Abstract section”.

Background

Line 69-73; can you provide some common examples for processed and ultra-processed foods so that readers can easily understand the differences between the two?

Answer: We are thankful to the editor for this comment. The examples have been provided in the” Introduction section” (Page 2, lines 69-70, and lines 73-74).

Methods:

line 141 and 150: it is not clear why the study considered primary school students as source population while the age group of interest is 10-19 years of age.

Answer: We are thankful the editor for this comment. We recognize this mistake. The text has already been rewritten correctly (Methods section- Study population, page 6, line 148).

Line 153-154: Sample size is calculated based on the assumption that the outcome of interested is altered lipid profiles in each district and this is not directly related with the current manuscript. How do you assure the adequacy of the sample size/power to address the objective of this specific manuscript?

Answer: We are thankful the editor for this comment. The sampling plan was defined for the study “Risk factors for cardiovascular disease among adolescents beneficiary of the Brazilian government National School Feeding Program (PNAE)”, which aimed to explore risk factors for cardiovascular diseases in adolescents. However, diet was included as one of the environmental variables of the study.

Line 189-90: the sentence “Recall surveys were performed at an interval of 30–45 days, according to the recommendation of Thompson and Byers” is not clear. Please provide additional description of this approach.

Answer: We would like to thank the editor for this comment. Since day-to-day variability in diet is high, the information from a single day cannot accurately reflect the usual diet of an individual. Other authors suggest collecting information on nonconsecutive days, since consecutive days suffer from the error in one day may be correlated with the error in the next day, and thus are not independent assessments1. Thus, we decided to apply the 24-hour dietary recall on 2 non-consecutive days, and set an interval of 30-45 days between the first and second 24-hour dietary recall collection. This information has already been rewritten for the purpose of clarification (Materials and Methods section, page 8, line 193-194).

1Frances E. Thompson, Amy F. Subar, Catherine M. Loria, Jill L. Reedy, Tom Baranowski Need for Technological Innovation in Dietary Assessment. J Am Diet Assoc. 2010 Jan; 110(1): 48–51. doi: 10.1016/j.jada.2009.10.008

Line 247: “Variables with p-values ≤ 0.30…….” can you please provide the list of variables considered for the bivariable analysis? As I stated above, I really question the importance of multivariable analysis here.

Answer: We appreciate the editor’s comment. The authors chose to use a relatively permissive p-value (0.3) to select candidate variables to be included in the multiple regression model to avoid the possible exclusion of important variables, based on an arbitrary statistical criterion1. Importantly, this p-value cutoff was only applied during the initial selection of variables to be included in the adjusted models. Standard p-value cutoffs (p = 0.05) were used for all final models. The variables considered for bivariable analysis are shown in Table 4 (vitamin A, B1, B2, B6, B12, folate, phosphorus, and selenium for processed foods, and vitamin B1, B2, B6, E, sodium, zinc, and iron for ultra-processed foods). 

1Martínez-González, M.A., Sánchez-Villegas, A., Fajardo, J.F. Bioestadística Amigable. 3. ed. Madrid: Diaz de Santos. 2014. 612 p.

Line 245-7: As I commented above, please also clearly indicate the dependent and independent variables of the model.

Answer: We are grateful to the editor for this comment. The purpose of the study is to evaluate the association between consumption of processed and ultra-processed foods with the prevalence of inadequate micronutrient intake. Considering the cross-sectional design of this study, one of the disadvantages is that the researcher does not know temporally what happens first, in this case, the inadequacy of nutrients or consumption of processed and ultra-processed foods. We examine only association and not cause-effect. Even if there is the assumption of this cause-effect relationship, there is no way to prove which occurred first. This methodological choice was made considering that consumption possibly does not interfere alone in only one micronutrient, and this effect needs to be controlled. Thus, with a cross-sectional design, the variables are placed interchangeably on different sides of the equation. Therefore, we clarify this information on Methods section (page 10, lines 249-250; lines 259-261).

Results

Please provide the operational definitions employed to classify the sexual maturation of adolescents.

Answer: We are thankful the editor for this remark. The Tanner scale is widely used in studies with adolescents to classify sexual maturation. This assessment was performed by pediatric endocrinologists with expertise in this field. This information is described in the “Methods section” (page 8, line 186-189).

Line 303: “high prevalence of pyridoxine intake” confusing, high or low intake?

Answer: We would like to thank the editor for this comment. The information has already been rewritten correctly (section Results, page 15, line 322-323).

Table 3: can you please explain why you limit yourself to the 15 micronutrients provided in the table?

Answer: We appreciate the editor’s comment. We decided to explore these 15 micronutrients, considering the nutritional importance of these nutrients for the development and health care of adolescents. In addition, we are more certain about the nutrition quality information in terms of the composition of the foods described in the tables.

97% more likely > almost two times more likely.

Answer: We are thankful to the editor for this comment. The information has already been rewritten as suggested (Results section, page 19, line 343).

Why it was not possible to present table 4 and 5 together in one table?

Answer: We would like to thank the editor for this question. The Tables 4 and 5 were presented together as suggested (Results section, page 20,21 and 22).

Table 4 and 5: Please clearly describe the dependent and independent variables, I also propose for removing the multivariable analysis.

Answer: We are thankful to the editor for this comment. To clarify the results, we included, in Table 4, which associations were related to the consumption of processed foods, and which were related to the consumption of ultra-processed foods.

Table 4 and 5: as long as p values are there I don’t see the purpose of having the Wald statistic there.

Answer: We are thankful to the editor for this comment. The Wald statistic is the most appropriate significance test in generalized (non-general) linear models for effect measures (PR or OR or RR). We decided to include the Wald statistic because the reader can confirm the statistical findings.

Others

Multivariate > Multivariable; bivariate > bivariable

Answer:

Reviewer 1

Processed and ultra-processed foods are associated with high prevalence of inadequate selenium intake and low prevalence of vitamin B1 and zinc inadequacy in adolescents from public schools.

The study addresses an interesting and relevant subject showing the association of consumption of processed and ultra-processed foods with micronutrient inadequacy in adolescents. However, the MS lacks logical presentation of information hence is difficult to understand.

The introduction doesn’t contain the relevant background information to guide readers to the rest of the manuscript. The justification or motivation to do this research is not well formulated.

Answer: We appreciate the reviewer’s comment. The introduction of the manuscript describes information about adolescents and the impact of food processing on health. The changes in adolescent eating patterns are presented in several countries, such as the United States, and others from Latin America, including Brazil. This information makes the reader interested in reading the manuscript. As suggested, we reinforce the motivation for conducting this research in the 

Introduction section (Page 5, lines 123-131).

The language not clear.

Answer: We would like to thank the reviewer for this comment. The manuscript was submitted for review of English language by a native speaker at Editage, as supported by the editing certificate provided below.

The outcome of this study mainly depends on the quality of the food composition table used to convert into nutrient intake of the present study subjects. However, this may not work for some nutrients such as selenium. This is because; selenium concentration in foods is mainly determined by the amount of the element in the soil which is quite variable in short distances. Thus, data from food composition is not reliable.

Answer: We are thankful to the reviewer for this comment. We are aware of the complexity of assessing selenium dietary intake. To minimize this problem, we use national tables. We checked the information on nutrients composition in special the amount of selenium in foods.

Parts of the methodology lack clarity and are not detailed. The authors in general tried to show the effect of level of processing on micronutrient intake inadequacy that consumption of processed foods was associated with lower intake of selenium but not Zn and B1. Processed foods in general are poor in micronutrient content. However, it is failed to identify whether foods in the present study were enriched with Zn and vitamin B1 concentration.

For example,

Line 48-50, Energy consumption from processed foods was associated with higher prevalence of inadequate selenium intake (p < 0.01) but lower prevalence of inadequate vitamin B1 intake (p = 0.04).

Line 50-54, Energy consumption from ultra-processed foods was associated with lower prevalence of inadequate zinc and vitamin B1 intake (p < 0.01 and p =0.03, respectively). An increase in the proportion of energy obtained from processed and ultra-processed foods may reflect higher prevalence of inadequate selenium intake and lower prevalence of vitamin B1 and zinc inadequacy.

Line 74-75. It is not always true to claim that sedentary behaviors among adolescents lead to greater consumption of packed or ready to eat foods.

Answer: We are thankful to the reviewer for this comment. Other authors have demonstrated the associations between sedentary behavior and consumption of ultra-processed foods. Costa et al (2015)1 observed that 40% of the schoolchildren reported daily consumption of at least one group of ultra-processed foods (39.7%; 95%CI: 39.2-40.3), while 68.1% (95%CI: 67.7-68.7) reported > 2 hours/day of sedentary behavior. Among schoolchildren with sedentary behavior > 2 hours/day, prevalence of daily consumption of ultra-processed foods was 42.8% (95%CI: 42.1-43.6%), higher than among those without a sedentary behavior (29.8%; 95%CI: 29.0-30.5%). Longer time spent in sedentary behavior was associated with higher prevalence of consumption of ultra-processed foods among Brazilian adolescents.

The information in the introduction section was rewritten for the purpose of clarification (Introduction section, page 3, line 75).

1COSTA, Caroline dos Santos et al. Sedentary behavior and consumption of ultra-processed foods by Brazilian adolescents: Brazilian National School Health Survey (PeNSE), 2015. Cad. Saúde Pública. 2018, vol.34, n.3. :e00021017 http://dx.doi.org/10.1590/0102-311x00021017.

Methodology

Line 136-140: Ethical consideration/approval has to be presented separately

Answer: We would like to thank the reviewer for this comment. We presented the information about ethical approval in a separated topic (Methods section, page 7, line 174-178).

Line 141-148: The inclusion and Inclusion criteria may work for the main study “Risk factors for cardiovascular disease among adolescent beneficiaries of the Brazilian government National School Lunch Program (PNAE)” but not for this specific MS.

The sample size was determined considering altered lipid profile as an outcome which is not consistent with the present study outcome.

Answer: We are thankful the editor for this comment. The sampling plan was defined for the study “Risk factors for cardiovascular disease among adolescents beneficiary of the Brazilian government National School Feeding Program (PNAE)”, which aimed to explore risk factors for cardiovascular diseases in adolescents. However, diet was included as one of the environmental variables of the study.

Weight and height measurement procedure is not detailed

Answer: We are thankful to the reviewer for this comment. Weight and height were performed at the school in duplicate by trained staff. The measurement procedures were carried out according to Habicht (1974)1. We chose not to describe the procedures in detail and inserted the reference used for standardization techniques1. The information was described in the Methods section, page 7, lines 182-184.

1Habicht JP. Standardization of quantitative epidemiological methods in the field. Bol Oficina Sanit Panam. 1974;76(5):375-384.

Line 179: ‘The morning was chosen to more easily enable methodological procedures’. This is not clear. Does this mean ‘anthropometric measurement was done during the morning? Why?

Answer: We would like to thank the reviewer for this comment. We recognize this mistake. The information was excluded from the text.

Line 182-185: It is not clear the importance of assessing pubertal stage given the age of study participants was known for this particular study.

Answer: We are grateful to the reviewer for this comment. Pubertal staging is important in studies with adolescents to correct variations in body composition, since puberty is a physiological process of hormonal maturation and somatic growth, which makes the organism able to reproduce. Changes in body composition, mediated mainly by the actions of sex hormones and growth, are characteristics of pubertal maturation and result in physical differences between sexes (Rogol, 2002)1. So as to make the research more complete, we decided to include this variable, describing the general characteristics of the population.

1Rogol AD, Roemmich JN, Clark PA. Growth at puberty. Journal of Adolescent Health. 2002;31:192-200. 

Reviewer 2

Introduction - it would be good to also have some systematic review to be cited to reiterate how the unhealthy dietary intake from low middle income countries/ Asian as well to show this is a global burden.

Answer: We would like to thank the reviewer for this comment. We have cited a systematic review about unhealthy dietary intake in Malaysia, a middle-income country in Asia (Introduction section, page 4, lines 94-96).

Methods - it is noted in methods the age of interest is 10-19. However, based on the results it appears only mean age 11.8 years. Why the age group seems to be specific (10-13)?

Answer: We are thankful to the reviewer for this comment. Data collection was performed with the students in the morning At this time of day, the age group of 10 to 13 years is more prevalent in schools in northeastern Brazil, and adolescents over 14 years of age are more prevalent in the evening period. This fact may explain the higher number of adolescents aged 10-13 years.

Explain why just use 24 hours diet recall and repeated twice. Is it the most suitable for adolescents studies? Since your interest is micronutrient and ultra processed food, which method of dietary intake should be more robust to be used for this adolescents group? The inadequate number of days for this might influence why you discovered such results.

Answer: We are thankful to the reviewer for this comment. As previously described, we followed the protocol of Thompson and Beyes, which establishes that for the evaluation of habitual food consumption, at least 2 recalls are required. In addition, we used the method AMPM To facilitate the accurate recall of dietary intake, AMPM employs five sequential passes in the interview: (1) asking the participant to start by quickly listing the foods consumed in the previous day (without the need for a time sequence); (2) asking about any foods they had forgotten to report in nine commonly forgotten categories of foods; (3) asking about the time and occasion of consumption for each food; (4) probing for specific details on foods, amounts consumed, and foods consumed between identified eating events; and (5) probing for whether any food that they had forgotten to report1.

1Moshfegh AJ1, Rhodes DG, Baer DJ, Murayi T, Clemens JC, Rumpler WV, Paul DR, Sebastian RS, Kuczynski KJ, Ingwersen LA, Staples RC, Cleveland LE. The US Department of Agriculture Automated Multiple-Pass Method reduces bias in the collection of energy intakes. Am J Clin Nutr. 2008 Aug; 88(2): 324–332.

What was the study response rate? How reliable is the reported Tanner staging? Any confirmation by the paediatrician?

Answer: We are thankful the reviewer for this comment. As described in the methodology, “A medical team of pediatric endocrinologists assessed pubertal staging. Sexual maturation was classified according to the Tanner scale into: pre-pubertal (stage 1), initial pubertal (stages 2 and 3), and final pubertal (stages 4 and 5) [24, 25].”. Thus, all adolescents evaluated by the doctor for sexual maturation were included in the study.

Results - The age group for 14 and above only 18? why is it so low? If this group is excluded, how different will the results be in Table 3?

Answer: We appreciate the reviewer’s comment. Data collection was performed with the students in the morning. In this period, the age group of 10 to 13 years is more prevalent in schools in northeastern Brazil, and adolescents over 14 years of age are more prevalent in the evening period. This fact may explain why most adolescents were aged 10-13 years. Even with a small number of adolescents over 14 years of age, we decided to maintain this age range, according to the inclusion criteria of the study. In Table 3, we decided to keep the group of adolescents aged 14-18 years, as the nutrient recommendations for this age range are different from those for the age range 10-13 years.

Figure - Out of 4457 subjects only 700+ enrolled. Why?

Answer: We would like to thank the reviewer for this comment. The sample size calculation indicated 483 students. However, 711 students were enrolled to compensate for the sample losses of 30%. The information has already been rewritten for the purpose of clarification (Methods section, page 6, line 152-156).

Since the two-stage random stratified sampling is used, why complex sample analysis not conducted?

Answer: We are grateful to the reviewer for this comment. We agree that a complex analysis with consideration of the sampling strategy could be an appropriate alternative to the data analysis we performed, considering that the sample was performed in two stages. However, in a thesis carried out during the study, the effect of the design was found to be less than 2.5% for all estimates, which did not significantly alter the estimates or the associated conclusions (Lyra, CO, Anthropometric Nutritional Status, Body Composition and blood pressure in adolescents [Thesis] Natal / RN: UFRN, 2012 https://repositorio.ufrn.br/jspui/handle/123456789/13265). As such, the authors chose not to perform the complex sampling analysis.

It appears highlighted micronutrient intake was low but do you have biomarkers to associate with?

Answer: We are thankful to the reviewer for this comment. The protocol defined for the study “Risk factors for cardiovascular disease among adolescents beneficiary of the Brazilian government National School Feeding Program (PNAE)” focuses on risk factors for cardiovascular diseases in adolescents. The diet was elected as the study variable for micronutrient intake assessment. In this study, biochemical biomarkers of micronutrients were not included.

Discussion - If ultra processed will lead to reduce certain micronutrient content, will enrichment with that vitamins in the food minimise the problem of suboptimal nutrient?

Answer: We are thankful to the reviewer for this comment. The enrichment with vitamins in the food can minimize the problem of suboptimal nutrient; however, excessive consumption of certain vitamins and minerals could have deleterious consequences on health and development of individuals and populations. Simultaneous micronutrient-delivery interventions could be challenging in terms of safety as the target populations may overlap, posing a risk of excessive intake of certain micronutrients1. Additionally, the bioavailability of micronutrients depends on food composition and possibly on its encapsulation method in some specific formulations2. This discussion is included in the manuscript (Discussion section, page 27, lines 473-480).

1Garcia-Casal MN, Mowson R, Rogers L1, Grajeda R; consultation working groups. Risk of excessive intake of vitamins and minerals delivered through public health interventions: objectives, results, conclusions of the meeting, and the way forward. Ann N Y Acad Sci. 2019 Jun;1446(1):5-20. doi: 10.1111/nyas.13975. Epub 2018 Oct 5.

2Monteiro CA, Cannon G, Moubarac JC, Levy RB, Louzada MLC, Jaime PC. The UN decade of nutrition, the NOVA food classification and the trouble with ultra-processing. Public Health Nutr. 2018; 21(1):5-17. doi: 10.1017/S1368980017000234.

Research has shown the association ultra-processed foods that contain added sugars, excess sodium, and unhealthful fats possibly lead to poor health outcome and is this pattern the same with yours?

Answer: We are thankful to the reviewer for this comment. Other authors concluded that there is higher caloric contribution of carbohydrates, lipids and saturated fatty acids of ultraprocessed foods, as well as a lower concentration of protein and fibers1. This pattern is also the same in Brazil, as demonstrated by Cunha et al (2018)2. This study confirmed that greater intake of ultra-processed foods is a marker of an unhealthy diet in Brazilian adolescents. Other studies confirmed the role of ultra-processed foods in the obesity epidemic among Brazilian adults and adolescents3.

1Monteles N, Larisse et al. The impact of consumption of ultra-processed foods on the nutritional status of adolescents. Rev. Chil. Nutr. 2019, vol.46, n.4, pp.429-435. http://dx.doi.org/10.4067/S0717-75182019000400429.

2Cunha DB, da Costa THM, da Veiga GV, Pereira RA, Sichieri R. Ultra-processed food consumption and adiposity trajectories in a Brazilian cohort of adolescents: ELANA study. Nutrition & Diabetes, 2018, v.8, n.28, p.1-9. DOI 10.1038/s41387-018-0043-z

3Louzada ML, Baraldi LG, Steele EM, Martins AP, Canella DS, Moubarac JC, Levy RB, Cannon G, Afshin A, Imamura F, Mozaffarian D, Monteiro CA. Consumption of ultra-processed foods and obesity in Brazilian adolescents and adults. Prev Med. 2015 Dec;81:9-15. doi: 10.1016/j.ypmed.2015.07.018. Epub 2015 Jul 29.

In the conclusion - "stress the importance of limiting personal consumption of processed and ultra-processed", how and where at school, home or institution and what is the allowed value? If there is any? What would be the most suitable take home message?

Answer: We are thankful to the reviewer for this comment. Healthy eating habits should be stimulated at home, at school or in institutions. The food guide for the Brazilian population1 recommends the consumption of fresh or minimally processed foods, in a large variety and predominantly of plant origin, because they are the basis for a nutritionally balanced, tasty, culturally appropriate diet for promotion of a socially and environmentally sustainable food system. This document recommends that the consumption of processed foods should be limited to small quantities, either as ingredients of culinary preparations, or as a meal complement. There is no recommendation for consumption of ultra-processed foods; the food guide for the Brazilian population recommends that people should avoid ultra-processed foods because of their ingredients – e.g., cookie sandwiches, packaged snacks, soft drinks and instant noodles. These products are nutritionally unbalanced. Because of their formulation and presentation, they tend to be consumed in excess and replace fresh or minimally processed foods. Their forms of production, distribution, commercialization and consumption adversely affect culture, social life and the environment.

1Brasil.Guia alimentar para a população brasileira.2nd ed. Brasília, Distrito Federal: Ministério da Saúde; 2014.

Reviewer 3

Abstract - Numerous sentences, such as the one highlighted below, starting with 'this':

"This was a cross-sectional study.”

Answer: We are thankful the reviewer for this comment. The sentences were rewritten according to your suggestion. (Abstract, page 2, lines 35-36).

-The term 'processed and ultra-processed foods’ is being used several times. It is very repetitive.

Answer: We are thankful the reviewer for this comment. We have edited the terms when appropriate. (Abstract, page 2, lines 34).

-Insert the acronym for 'estimated average requirement

Answer: We are thankful to the reviewer for this comment. The acronyms for 'estimated average requirement’ was added (Abstract, page 2, line 41).

Introduction

-The information stated in lines 96 and 97 has already been said in the introduction’s section elsewhere.

Answer: We are thankful to the reviewer for this comment. We reformulated the information in lines 96 and 97, and in the last paragraph of the introduction. (Introduction section, page 4, lines 96-99; page 5, lines 123-131).

-The paper discusses that some results found might be due to differences of eating habits or economic status of northeastern Brazil, in comparison to the rest of the country, but it does not mention any of this in the introduction. The authors need to expand on this.

Answer: We would like to thank the reviewer for this comment. We included information about eating habits or economic status of northeastern Brazil (Introduction section, page 5, lines 119-122).

Methods

-Were rural schools included? The access to milk and dairy products could be different compared to the population from the urban ones.

Answer: We are thankful to the reviewer for this comment. The schools included in this study are in the urban area of Natal city, northeastern Brazil.

-In Fig 1, why are there 700 adolescents enrolled when it is mentioned above only 11 were transferred from school?

Answer: We are thankful to the reviewer for this comment. We recognize this mistake. At first, 711 were enrolled instead of 771. Then, 11 were transferred from school, and 700 remained effectively enrolled. The information has already been rewritten correctly in Figure 1.

-What about those participants who were taking dietary supplements? Was this type of information collected or evaluated?

Answer: We are thankful to the reviewer for this comment. The information about the use of supplement was collected, but we did not find any adolescent using these products.

-Was the apportioning and quantification of food reviewed by the 24-hour dietary recalls' interviewers?

Answer: We are thankful to the reviewer for this comment. All the 24-hour dietary recalls were reviewed and the database was checked by more than one researcher.

Discussion

-"We observed low percentages of energy intake from ready-to-eat and semi-ready-to- eat meals and sweetened milk drinks from the ultra-processed foods group.” I would prefer the passive voice.

Answer: We thank the reviewer for this comment. The sentence was rewritten according to your suggestion. (Discussion section, page 23, lines 393).

-"We observed low percentages of energy intake from ready-to-eat and semi-ready-to- eat meals and sweetened milk drinks from the ultra-processed foods group. This finding could be attributed to the fact that most families living in northeastern Brazil are characterized as low-income families [39, 40], as an increase in the intake of ready-to-eat and semi-ready-to- eat meals and sweetened milk drinks has been associated with an increase in family income per capita [40].” Is it really different from "cakes, pies, and cookies"?

Answer: We are thankful to the reviewer for this question. In northeastern Brazil most of the cakes, pies, and cookies consumed are non-industrialized preparations, and therefore, they differ from the ready-to-eat and semi-ready-to-eat meals and sweetened milk drinks. The fact that they are homemade preparations, and therefore cheaper, does not necessarily make the food healthier, because ultra-processed culinary ingredients are included in their preparation.

-“In our study, we observed low intake of foods that are significant sources of selenium, such as fish, whole grains, and Brazil nuts.” Can this information be drawn from table 2?

Answer: We are thankful to the reviewer for this comment. This information cannot be drawn from Table 2, because the foods cited as selenium sources are not part of the processed and ultra-processed food group. This information was collected from the 24-hour dietary recall. 

-“Brazilian children aged 2–3 years typically have a low prevalence of iron, vitamin C, vitamin A, calcium, and folate inadequacy. None of these children had intakes less than the EAR value for zinc. Fortified foods contributed to micronutrient supply. However, 4·0% of children exceeded the UL for vitamin A, 3.1% for zinc, 1.1% for folic acid, and 0.2% for iron. These results suggest a low prevalence of inadequate micronutrient intake among children, with the implication that this group could be at risk of excessive micronutrient intake provided by ultra-processed foods [59].” Isn't this paragraph deviating from the population targeted at this paper?

Answer: We are thankful to the reviewer for this comment. Although the population is different from the one addressed in the study, we decided to highlight this information because it is one of the few Brazilian studies that has addressed this theme. We rewrote the paragraph to improve the interpretation (Discussion section, pages 26-27, lines 473-480).

-According to results presented, processed and ultra-processed foods are associated with a low prevalence of vitamin B1 and zinc inadequacy. The authors need to expand on this.

Answer: We are thankful to the reviewer for this comment. The explanation about this topic is in the Discussion section, page 27, lines 461-472.

-What are the limitations of the 24-hour dietary recall?

Answer: We are thankful to the reviewer for this comment. The main challenges to accuracy with the 24HR are attention and memory. Many respondents are challenged with distinguishing between what they usually eat and what they ate yesterday, opening the possibility for omissions and intrusions (foods reported, but not actually eaten). Actual memory of distinct events decays with time, which has been documented to start within an hour of the meal. Portion size estimation is also challenging as the amount consumed has to be both remembered and accurately estimated. Since day-to-day variability in diet is high, the information from a single day cannot accurately reflect the usual diet of an individual. Multiple days of information are collected and statistical techniques employed to address this problem1.

1Frances E. Thompson, Amy F. Subar, Catherine M. Loria, Jill L. Reedy, Tom Baranowski Need for Technological Innovation in Dietary Assessment. J Am Diet Assoc. 2010 Jan; 110(1): 48–51. doi: 10.1016/j.jada.2009.10.008

-The results need to be further compared with the literature.

Answer: We are thankful to the reviewer for this comment. The results are discussed on pages 23-25, lines 378-597.

---

## [Decision Letter · Decision Letter 1]

28 Oct 2019

Processed and ultra-processed foods are associated with high prevalence of inadequate selenium intake and low prevalence of vitamin B1 and zinc inadequacy in adolescents from public schools in an urban area of northeastern Brazil

PONE-D-19-18574R1

Dear Dr. Sena-Evangelista,

We are pleased to inform you that your manuscript has been judged scientifically suitable for publication and will be formally accepted for publication once it complies with all outstanding technical requirements.

With kind regards,

Samson Gebremedhin, PhD

Academic Editor

PLOS ONE

Reviewer's Responses to Questions

**Comments to the Author**

1. If the authors have adequately addressed your comments raised in a previous round of review and you feel that this manuscript is now acceptable for publication, you may indicate that here to bypass the “Comments to the Author” section, enter your conflict of interest statement in the “Confidential to Editor” section, and submit your "Accept" recommendation.

Reviewer #3: All comments have been addressed

2. Is the manuscript technically sound, and do the data support the conclusions?

Reviewer #3: Yes

3. Has the statistical analysis been performed appropriately and rigorously? 

Reviewer #3: Yes

4. Have the authors made all data underlying the findings in their manuscript fully available?

Reviewer #3: Yes

5. Is the manuscript presented in an intelligible fashion and written in standard English?

Reviewer #3: Yes

6. Review Comments to the Author

Reviewer #3: (No Response)

7. PLOS authors have the option to publish the peer review history of their article (what does this mean?). If published, this will include your full peer review and any attached files.

Reviewer #3: No

---

## [Editor Report · Acceptance letter]

11 Nov 2019

PONE-D-19-18574R1 

Processed and ultra-processed foods are associated with high prevalence of inadequate selenium intake and low prevalence of vitamin B1 and zinc inadequacy in adolescents from public schools in an urban area of northeastern Brazil 

Dear Dr. Sena-Evangelista:

I am pleased to inform you that your manuscript has been deemed suitable for publication in PLOS ONE. Congratulations! Your manuscript is now with our production department. 

With kind regards,

on behalf of

Dr. Samson Gebremedhin 

Academic Editor

PLOS ONE